# Detection of Tram Wheel Faults Using MEMS-Based Sensors

**DOI:** 10.3390/s22176373

**Published:** 2022-08-24

**Authors:** Yohanis Dabesa Jelila, Wiesław Pamuła

**Affiliations:** 1Department of Transport Systems, Traffic Engineering and Logistics, Faculty of Transport and Aviation Engineering, Silesian University of Technology, 40-019 Katowice, Poland; 2Faculty of Mechanical Engineering, Jimma Institute of Technology, Jimma University, Jimma P.O. Box 378, Ethiopia

**Keywords:** MEMS, faulty wheel detection, MODWPT transforms, tram, discrete optimization problem, DW function, rolling wheel data

## Abstract

Micro-electromechanical-systems (MEMS) based sensors are used for monitoring the state of machines in condition-based maintenance tasks. This approach is applied at tram depots for the purpose of identifying faulty wheels on trams in order to eliminate defective trams at the entry or dispatch gates. The application of MEMS-based sensors for the detection of wheel faults is the focus of this study. A method for processing of the collected sensor data is developed. It is based on assessing the energy of vibrations at different frequency bands. Maximal Overlap Discrete Wavelet Packet Transform (MODWPT) is used for obtaining a description of the sensor data. The task of finding the energy threshold for detecting faulty wheels, frequency band and parameters of MODWPT which most distinctly distinguish the wheels is the goal of the method. The weighted difference (DW) between the extreme values of energy in a frequency band for normal and faulty wheels is proposed as the measure of the ability to distinguish the wheels. The search for the solution is formulated as a discrete optimisation problem of maximising this measure. Both the simulation and experimental results indicate that faulty wheels have greater vibration energy than normal wheels. The properties of this approach are discussed and evaluated.

## 1. Introduction

Potential factors that determine the success of tram transport are the durability, reliability, and safety of tram wheel-sets [1,2,3]. The wheel-set (wheel, axle, and axle box) of a tram vehicle is a critical component that contributes significantly to the vehicle’s stability while it is in motion by ensuring ride quality, transmitting traction, and braking force to the track. Thus, tram vehicle safety is primarily determined by the health of its wheel-sets. The problem of detecting faulty wheels is of paramount importance for safeguarding the safety of tram operations.

Unlike trains, city trams ride on very sharp curves, turnouts, rail joints, and high-traffic intersections, allowing them to efficiently utilize the space available in the city. The operation of a tramway in such a dynamic environment causes the most significant operational problems, such as the wear of wheels and rail irregularities when a tram passes [4]. In addition, the continuous movement of tram vehicles on such a dynamic rail track causes the physical deterioration of the wheel–rail contact surface. Moreover, the tramway crosses highway roads at multiple intersections, requiring frequent hard braking. This, in turn, causes wheel locking, excessive deformation, noise, unnecessary vibration, and high heat dissipation at the wheel–rail contact areas [5].

Due to these dynamic operational conditions, tram wheels are subjected to cyclic impact forces resulting in various defects that impair their rotational smoothness. Wheel flatness, wear, eccentricity, discrete defects, roughness, spalling, and shelling are among the most significant categories of defects [6]. Wheel-flat faults are the most common localized defect and give rise to other families of wheel damage. They result in the course of wheel–rail interaction during hard braking or slipping caused by debris on the rail [7].

Wheel-rail maintenance workers and managers strive to identify problems with wheels to avoid costly system-wide repairs. The wheel-assembly’s state should be monitored and maintained effectively before huge damage can occur. Early wheel-fault detection is required to ensure tram safety and stability. Therefore, fault detection and condition-based maintenance (CBM) are a profitable strategy for railway assets. This strategy can assist in ensuring increased protection and serviceability by identifying component problems as early as possible [8].

CBM can extend component lifetimes by minimizing downtime and enables operators to optimize any replaceable component in a particular operational condition by altering the dependency on scheduled maintenance and ensuring a unique component repair schedule [9]. Tram-wheel condition monitoring and fault diagnostic techniques are widely practiced in the railway industry, with applications of predictive and condition-based maintenance.

Currently, there are two broad categories of wheel-fault detection methods. These are wayside and on-board detection techniques. The deployment of either wayside or on-board detection depends on type of the resource we are going to diagnose and types of rail transport. 

On-board detection approaches are methods employed on the train to detect the nature of the fixed assets, such as switches and rail conditions, whereas the wayside-mounted detection approaches are applied to study the health condition of the moving components on the train such as wheels, axles, and bearing elements [10,11]. On-board detection is not the focus of this study due to its drawbacks: large data preparation, tedious calibration, and specific wheel design requirements for different vehicles.

The wayside detection of wheel faults is based on measuring the effects of wheel–rail interaction during the vehicle’s passage by placing a sensor unit at a single point. This method is inexpensive, and it overcomes the drawback of on-board detection techniques. The changes of acting forces, accelerations, or deflections are measured using strain gauges, accelerometer sensors, temperature sensors, mechanical sensors, and vibration- and noise-based sensors. These diagnostic devices have digital outputs, and the continuous time signals collected are digitised and treated as discrete-time-series signals.

These discrete-time-series signals are mostly processed using techniques in time, frequency, and joint time–frequency domains to reveal the conditions of the wheels. The frequency bandwidth changes of such signals fall within a few kHz. This limited range of changes opens the field for the application of MEMS-based sensors. Commonly available MEMS devices can sample accelerations up to few thousand times a second. The ranges of sensed acceleration values exceed hundreds of m/s^2^ [12,13,14]. For example, IIS2DH produced by STMicroelectronics has a sampling range up to 5.3 kHz and +/− 16 g sensing scale. Similar parameters characterise the series ADXL3xx sensors in the production portfolio of analogue devices.

The application of MEMS-based sensors for the detection of wheel faults is the focus of this study. To achieve the objective of the study, we have developed a method for identifying defective wheels that detects faults within a frequency band limit. This identifies faulty wheels and suggests maintenance and replacement.

The remaining parts of the paper are organized as follows. Section 2 presents the background of wheel-fault detection methods. The wheel-faults detection method, MODWPT properties evaluation and discrete optimisation problem formulation is described in Section 3. Solution search limits, a case study of tram depot test and validation of results using the proposed method for tram depot maintenance work are discussed in the following section. The concluding section summaries the characteristics of the method and proposes further development directions.

## 2. Related Works

Wheel-fault diagnostics have been the subject of extensive study over the past three decades, and numerous techniques have been developed. For instance, Bosso et al. [15] proposed an onboard wheel-flat diagnostic and detection algorithm. The algorithm was based on the calculation of a wheel-flat index number and enables the early detection of the presence of flats and the quantification of the severity of the problem based on the wheel-flat index number. By employing data from vertical axle-box vibration acceleration sensors, Sun et al. [16] developed a detection framework, based on the angle domain synchronous averaging technique, for monitoring the condition of railway wheels. This approach mitigates the influence of background noise. A systematic literature review of condition monitoring of rail transport systems based on bibliometric performance analysis was conducted by Kostrzewski and Melnik [17]. The authors describe the current trends of condition monitoring approaches and their role in maintaining rail transport systems over the last decades. In addition, the authors predict future research directions and perspectives in the condition monitoring of rail transport systems.

Polygonised railway wheel detection based on the numerical time-frequency analysis of axle-box acceleration is suggested by Song et al. [18]. The authors assess both the development of wheel polygonization and its degree by measuring the acceleration of the axle box. However, in this work, only computational simulation is reported. In the same manner, Wang et al. [19] also devise a new discrete Fourier transform (DFT)-based dynamic detection framework for the polygonal wear state of railway wheels.

Turabimana et al. [20] designed an onboard measuring unit for wheel flange faults in railway vehicles using an inductive displacement sensor. The unit works in both static and dynamic states of the vehicle and records the wheel-flange thickness data to identify the wheel fault. However, the work does not present the time-frequency response of each wheel, and it is difficult to differentiate the fault condition of the wheel element. 

Chen et al. [21] introduced hybrid microphone-array signal processing to identify faulty wheels and estimate ground impedance. Mosleh et al. [22] used an envelope spectrum analysis approach for flat-wheel detection in railway train wheels. Scalea and McNamara [23] applied longitudinal and lateral transient vibration characterization of railroad tracks using wavelet transforms. In [24], Brizuela et al. report an ultrasound technique based on measuring the changes in the round-trip time of flight (RTOF) of the ultrasound pulse to the rail–wheel contact point for detecting and quantifying wheel flats. 

Madejski and Gola [25] presented a tram-wheel geometry monitoring system employing an array of accelerometers mounted on rail tracks. The amplitude and frequency changes of the signals are tracked. Barman and Hazarika [26] investigated an accelerometer-based system that detects and identifies the faults of a train using linear time-frequency analysis. The Wigner–Vile transform of the vibration signal during the movement of a train over the track is the basis for analysis and fault detection. However, all studies conducted using frequency methods lack time information and the impulsive loading event of the fault condition is not clearly known as a function of time.

Gao et al. [27], proposed a flat-wheel detection and quantification method based on measuring the wheel–rail impact force of the entire wheel circumference. Two reflective optical position sensors, mounted along the rail, are used to detect the displacement. Impact-response curves indicate the condition of the wheel and enable the determination of the parameters of wheel flatness. 

Numerous time-frequency analysis techniques have been employed to diagnose wheel faults, including short-time Fourier transforms (STFTs), wavelet transform (WT), the Hilbert Huang transform (HHT), and empirical mode decomposition (EMD). In [28], Nowakowski et al. reported envelope analysis with Hilbert’s transform to detect tram-wheel flat spots and compare the time-frequency vibration signature analysis of tram wheels; however, a wheel with a flat spot was not clearly identified. The authors regard the vehicle accelerations as a basic physical quantity that determines the fault condition of the wheel.

In [29], Komorski et al. presented wheel-flat detection using advanced acoustic signal analysis techniques. The authors record the acoustics of tram-wheel fault conditions using three arrays of microphones mounted on the tramway and apply the Hilbert transform and spectrum envelope analysis for detecting faults. According to the results of the tests, the acoustic signal recorded near to a passing rail vehicle is an excellent carrier of diagnostic information for the detection of flat wheels. On the other hand, acoustic signals near to a passing rail vehicle can be highly affected by other sound signals and further acoustic signal filtration is required.

Yue Jianhai et al. [30] used continuous wavelet transform to detect meshing irregularities in wheels with tread defects. The sampling frequency of 10 kHz is chosen for a vehicle speed of 60 km/h. However, the applied method detects wheel flats better than treads, and the signal response of a flat wheel has the highest amplitude. Ghosh et al. [31] report the use of fast Fourier and wavelet transforms to detect the condition of rail tracks in real time. Results show that corrugation faults are more likely to be detected by fast Fourier transform (FFT) than WT, while crack damage is more likely to be detected by WT than FFT. In this study, vibrations are measured using accelerometer sensors mounted on the axle boxes of service trains, which is susceptible to interference from other vibration sources. The suggested method is less suitable for detecting wheel faults.

Zhang et al. [32] established an adaptive parameter blind source separation (BSS) approach for the diagnosis and monitoring of wheel defects. Signals from acceleration sensors mounted on wheels are processed. BSS can separate weak fault sources from a mixture of vibration signals related to train movement. A similar approach for the detection of wheel thread defects is presented in [33]. Wayside-mounted fibre Bragg gratings (FBGs) are used for measuring rail deflections. Defect-sensitive features of the measurements are obtained using Bayesian blind source separation. However, numerous variables, such as train speed variation, sharp rail geometric variation and the location of the FBG sensor with respect to sleepers influence defect-detection results.

Li et al. [34] proposed an adaptive multi-scale morphological filter for the fault detection of railway wheel flats. The authors report simulation results using simple vibration models consisting of one impulsive function and two sine-cosine harmonic waves with Gaussian white noise. Results confirm the detection capability of the filters, but no field investigation data are provided. An adaptive chirp mode decomposition method for railway wheel-flat detection under variable-speed conditions is proposed by S. Chen et al. [35]. The authors apply a time-frequency analysis technique to precisely extract the time-varying fault characteristic frequencies and thus successfully detect the faults.

Ye et al. [36] proposed a data-driven method for detecting flat wheel lengths. The authors use a multibody dynamics model (MBS) to create artificial axle-box acceleration data for a 20 mm wheel flat at variable vehicle speeds. A Kriging surrogate model is used to model axle-box acceleration and particle-swarm optimisation is used to calculate the wheel-flat length. However, the accuracy of this technique depends on the MBS model, the wheel-flat mode and feature-selection method, and more features that reflect the real-time status of wheel-flats in real conditions are required.

The reported methods for detecting wheel faults are based on mature technologies for measuring accelerations. It is crucial to detect wheel faults under variable conditions with strong interference that corrupts the fault-signal characteristics. The collected data are processed using a number of frequency-analysis techniques. These approaches successfully detect faults but can be difficult or costly for field implementation. 

MEMS-based sensors and MODWPT are proposed for evaluating the energy of vibration in different frequency bands. The value of vibration energy in characteristic frequency bands describes the condition of the wheels.

The contributions of this paper are:Demonstration of the capability of using MEMS vibration sensors with limited frequency bandwidth for detecting tram-wheel faults.Development of a tram-wheel fault-detection method using MODWPT for processing MEMS sensor data.Substantiation of the idea that vibration-energy differences in characteristic frequency bands are adequate for distinguishing faulty and normal tram wheels.

In this study, a method for processing the collected sensor data is developed. It is based on assessing the energy of vibrations in different frequency bands. Maximal overlap discrete wavelet transform (MODWT) is used for obtaining a description of the sensor data. The properties of this approach are discussed and evaluated.

## 3. Methods for Detection of Tram-Wheel Faults

Trams, in comparison to trains, are much lighter vehicles and this strengthens the idea of using MEMS-based sensors for the detection of wheel faults. The measurement properties of these sensors limit the detection capability.

The research problem, expressed as a question, is as follows: Is it possible to detect tram-wheel faults using MEMS-based sensors?

The implementation of such a detection method can become a routine maintenance task at tram depots. It can be applied for monitoring the conditions of incoming and outgoing trams. This implies that the vehicles travel at low speeds and in consequence their wheels interact with rails with much lower dynamic forces. The problem of collecting sensor data may be highly hindered by the condition of rail tracks at the depot.

The solution of this problem requires the development of a method for extracting sensor-data features disrupted by several obscuring factors. Wheel impacts at rail joints, loose sleepers, and damaged rails are the most significant sources of interference. Such interference sources can be modelled using random processes with changing parameters. The random nature of the interference demands a careful approach to the extraction task in order to avoid the erroneous interpretation of sensor data. The analysis of the variation in energies of the sensor data is proposed as the basis for the method of wheel-fault detection and wavelet transform is a promising tool for sensor-data feature extraction. 

Wavelet transforms have proven useful for the analysis of sensor-data samples. The transform coefficients are wavelet and scaling, at different levels of decomposition, which represent properties of the samples related to time and changes of value [37]. The discrete wavelet transform (DWT) decomposes the scaling coefficients at each level based on a pyramidal algorithm [38] and this leads to some loss of description capability [39,40,41,42]. The wavelet packet transform (WPT) decomposes both the wavelet and scaling coefficients. The transforms are not translation invariant, that is, a shift of the start of processing samples gives a different set of coefficients. This property is not desirable when processing sensor data. 

### 3.1. Maximal Overlap Discrete Wavelet Packets Transforms

Maximal overlap discrete wavelet transform (MODWT) overcomes the lack of translation-invariance; it is invariant to circular shifting of the data samples. Maximal overlap discrete wavelet packet transform is developed on the basis of MODWT and includes the decompositions of the wavelet coefficients. MODWPT is an energy-conserving transform, that is, the sum of the total energies of the MODWPT coefficients is equivalent to the energy of the signal [43].

The following research hypothesis is proposed as the basis of the study: vibration energy, calculated using MODWPT coefficients, in determined bands of frequencies, enable the detection of wheel faults.

The set of N sensor samples $S=\left\{s_{0},s_{1},\;\ldots ,\;s_{N-1}\right\}$ is decomposed using a pair of low- and high-pass filters *h_l_* and *g_l_* [44]. The low-pass scaling filter is defined as: $\left\{g_{l}:l=0,\;\ldots ,\;L-1\right\}$ whereas the high-pass wavelet filter is $\left\{h_{l}:l=0,\;\ldots ,\;L-1\right\}$, where L is the length of filter and L≤N. The filters are scaled by the factor $1/\sqrt{2}$ to satisfy the energy conservation constraint. The MODWPT coefficients are efficiently calculated using a recursive process. At the first level, S is filtered using *h_l_* and *g_l_* to obtain the first-level coefficients $W_{1,0}=\left\{W_{1,0,k}\;:k=0,1,\;\ldots ,\;N-1\right\}$ and $W_{1,1}=\left\{W_{1,1,k}\;:k=0,1,\;\ldots ,\;N-1\right\}$. Next, the filters at level *j* are extended by adding $2^{j-1}-1$ zeros, for j≥1, between the filter coefficients ($g_{l},h_{l}$). and the calculation is repeated to obtain the next-level wavelet coefficients $W_{2,0},W_{2,1},W_{2,2},W_{2,3}$. The process is continued until the desired level of decomposition is reached. 

The MODWPT coefficients at node (*i*, *j*) are given by [45]:(1)$$W_{i,j,k}=\overset{L-1}{\underset{l=0}{\sum }}f_{j,l}W_{i-1,\left[\frac{j}{2}\right],\left(k-2^{\left(i-1\right)}l\right)\;mod\;N}$$
where
(2)$$f_{i,l}=\{\begin{array}{c} g_{l}\;\;\;\;\;\;\;if\;n\;mod\;4=0\;or\;3\\ h_{l}\;\;\;\;\;\;if\;n\;mod\;4=1\;or\;2 \end{array}$$

*n* is the frequency band number at the decomposition level *i*. There are 2^i−1^ frequency bands at each decomposition level; ‘mod’ means modulus after division.

The energy of the collected sensor data is calculated as:(3)$${\left\|S\right\|}^{2}=\overset{N-1}{\underset{n=0}{\sum }}s_{n}^{2}$$

Expressed using MODWPT coefficients at decomposition level *i*:(4)$${\left\|S\right\|}^{2}=\overset{2^{i}-1}{\underset{n=0}{\sum }}\|W_{i,n}\|{}^{2}$$

This is the sum of energies in frequency bands of the sensor samples. The sample rate *f_s_* of the sensor determines the frequency ranges of the bands $b_{n}^{i}$ at decomposition level *i*:(5)$$b_{n}^{i}=\{n(f_{s}/2^{i}),\left(n+1)(f_{s}/2^{i}\right)\},$$

Figure 1 shows an example of MEMS sensor data collected at a tram depot test site. The accelerations of the rail are measured during the passage of the tram, travelling at about 2 m/s. The sampling rate *f_s_* is set to 1 kHz. 

The sensor registers vibrations in the rail caused by the rolling tram wheels—Figure 1b—and much higher vibrations caused by impacts of the wheels at rail joints—Figure 1a. The rail segments at the depot are not welded. The impact vibrations strongly mask the effects of wheel faults which also cause impact-like vibrations in the rails. Impacts at rail joints can randomly change, thus introducing unpredictable inference. The collected sensor dataset is cut into impacts and rolling data. The rolling data are chosen for the detection of wheel faults. The lengths of rolling periods enable the registration of several thousand samples which are not corrupted by impacts. The image of the sensor data does not directly indicate the wheel conditions. Wheel faults cause impact-like vibrations with a specific frequency spectrum dependent on the sort and size of the fault. Vibration energy is proposed as the measure of the properties of the wheel movement [46].

Figure 2 presents an incidence of the relative energy variation in registered vibrations of the rails during tram movement. The values are obtained using MODWPT coefficients and a preliminary set of sensor samples, as noted earlier.

There are energy peaks in the bands 150–230 and 400–460 Hz. The superimposed graphs of normal and faulty wheels show common behaviour. Further analysis is required to extract frequency bands, which differentiate the vibrations caused by the movement of the wheels.

### 3.2. Discrete Problem Optimisation

The important optimization criterion is the complexity of calculations as the signals from MEMS sensors are usually processed using microcontrollers. The criterion is additionally constrained by the requirements of efficient maintenance practices and parameters of the available MEMS vibration sensors. The description properties of the MODWPT coefficients are determined by the wavelet type and the level of decomposition. Daubechies wavelets, Coiflets, and Symlets are candidates for use. The level of decomposition defines the frequency resolution of the analysis.

The task of finding the energy frequency band and parameters of MODWPT which most distinctly distinguish the wheels is the goal of the study. This can be formulated by a discrete optimization problem as the variables are discrete. The weighted difference (DW) between extreme values of energy in the band for the normal and faulty wheels is proposed as the measure of the ability to distinguish the wheels. The DW values are positive when the collected sets of samples can be differentiated. The search is limited by the set of wavelet types, number of samples processed, and the range of decomposition levels. The width of the frequency band may be widened to take into account the behaviour of the signal in neighbouring bands. The optimization problems and resulting search for a solution has five parameters which depend on the condition of the rails at the depot, the kind and size of the wheel faults which require maintenance, and the speed of travelling of the tram. The optimization goal is to find the largest difference of energies in a frequency band between normal and faulty wheels, measured in tests at the tram depot. The tests represent routine procedures of tram dispatch and arrivals.

The objective function is:(6)$$\max DW\left(w,N,i,k,n\right)=\frac{\underset{m}{\min}E_{f}\left(n(f_{s}/2^{i}\right),\left(n+k)(f_{s}/2^{i}\right))-\underset{m}{\max}E_{n}\left(n(f_{s}/2^{i}\right),\left(n+k)(f_{s}/2^{i}\right))}{\underset{m}{\max}E_{n}\left(n(f_{s}/2^{i}\right),\left(n+k)(f_{s}/2^{i}\right))}$$
expressed using wavelet coefficients:(7)$$\max DW\left(w,N,i,k,n\right)=\frac{\underset{m}{\min}\sum_{j=n}^{n+k}\|W_{i,j{}_{f}}{\|}^{2}-\underset{m}{\max}\sum_{j=n}^{n+k}\|W_{i,j{}_{n}}{\|}^{2}}{\underset{m}{\max}\sum_{j=n}^{n+k}\|W_{i,j{}_{n}}{\|}^{2}}$$
where

m—number of tests;

N—number of acceleration samples collected in a test;

E—energy of the test samples in a frequency band (normal/faulty wheels).

It is subject to constraints:wavelet types: w = {Db, coif, sym};number of samples processed: N = {2000, 3000, …, 9000};number of decomposition levels: $i=\left\{1,\;\ldots ,log_{2}\left(N\right)\right\}$;number of combined frequency bands: k = {0, …, N/2};starting number of the frequency band: n = {0, …, N}.

The solution defines the parameters of calculating the MODWPT using a set of samples from a MEMS-based acceleration sensor. The mid value between $\underset{m}{\min}E_{f}$ and $\underset{m}{\max}E_{n}$ in the obtained frequency band is the threshold for detecting faulty wheels. 

The threshold for detecting faulty tram wheels is:(8)$$\mathrm{TH}=\underset{m}{[\min}E_{f}\left(nf_{s}/2^{i}\right)+\underset{m}{\max}E_{n}\left(nf_{s}/2^{i}\right)]/2$$

The energy of samples in the frequency band above the threshold (TH) signals a faulty wheel.

The number of samples processed determines the length of time for collecting the samples. It is limited by the time between consecutive impacts of the tram wheels. The time between impacts in turn is determined by the speed of the moving tram and length of the rail track between joints. In the case of tram-depot maneuvers, the number amounts to several thousand samples when the sensors’ sampling rate equals 1 kHz. 

### 3.3. Method for Detecting Tram Wheel Faults

The proposed detection method consists of the following steps: Registration of rail-vibration data using a MEMS sensor with a frequency bandwidth of not less than 1 kHz.Determination of the DW parameters: w,N,i,k,n using the collected vibration data.Calculation of the detection threshold value *TH*.Calculation of wheel energy in the determined frequency band (2) and classification of the wheel condition using the threshold (3).

The resultant detection threshold TH is determined by the state of the rails at the tram depot, which defines the vibration image registered by the MEMS sensor. This implies that when the method is implemented, it may require updates when the rail conditions at the depot change.

## 4. Validation of the Method and Discussion

To limit the scope of the search for the solution, the ranges of possible values of the DW variables are restricted. The DW value changes are analysed to determine the directions for controlling this process. Crucial for the resolution of the frequency analysis, apart from the sampling rate, is the length of the sequence of MEMS-sensor samples and thus the maximal decomposition level. Preliminary measurements show that most of the signal energy is gathered in two narrow frequency bands. The highest frequency of the bands does not exceed 450 Hz which indicates that the chosen 1 kHz sampling rate is sufficient to obtain information on the wheel state. 

### 4.1. Solution Search Limits

In the first step, for limiting the search for the solution, the frequency resolution is restricted to 1 Hz. This value is common for describing vibration properties of objects. This assumption determines the minimal number of samples $N=2^{\left\lceil log_{2}\left(f_{s}/2\right)\right\rceil }$, that is, 512 and maximal decomposition level *i* = 9. In this case, using MODWPT, the frequency bands are described using single coefficients which can be sensitive to signal disruptions. In order to diminish the influence of possible interference, the set of samples is extended to 4096, ensuring 8 coefficients for the description of the frequency bands. The set of 4096 samples is collected in about 4 seconds which is acceptable for performing wheel diagnosis in real time. Extending the set of samples may be inadvisable because the speed of moving tram is variable, and a larger set may include the impact data. Additionally, the idea of combining the vibration energy of neighbouring frequency bands is abandoned. The first step reduces the number of variables of the DW function to: *w*—wavelet type, *i*—the level of decomposition, *n*—starting number of the frequency band. The starting number of the frequency band indicates the range of frequencies of the band. This enables a graphical analysis of the DW behaviour.

Figure 3 presents DW values in [%] using wavelets with three vanishing moments. Only the positive values are shown. Positive values enable the calculation of the detection threshold. The largest values are obtained for the 9th level of decomposition using db3 and sym3, that is, the width of the frequency band amounts to about 1 Hz. At the 7th level of decomposition, all the graphs indicate in a consistent manner, that the differences in vibration energy in the frequency band 418–422 Hz are significant for the detection of faulty tram wheels in the tram depot where the tests were carried out.

Table 1 lists DW values for representative wavelets and decomposition levels 5–9. 

DW values for Daubechies wavelets and Symlets are similar, which is consistent with the results presented in [47]. Coiflets at the 7th level of decomposition give consistent high DW values which indicates favourable detection properties. The authors of [48] prove that Coiflets can be more effective than Daubechies wavelets for describing object properties.

DW values obtained for the lower (less than 6th) levels of decomposition are insignificant and the variation in vibration energy is very minimal for the three wavelet families presented in Table 1. Therefore, at lower levels of decomposition, the wheel-fault detection is difficult.

In the final step, for limiting the search for the solution, the decomposition level is set to 7, although the 9th level gives the highest values. The 7th level of decomposition gives wider frequency bands; in this case, the width is about 4 Hz. Wider frequency bands give a more robust approximation of wheel-fault properties. Coiflets are chosen as the MODWPT calculation basis.

Increasing the number of wavelet vanishing moments—length *L* of the wavelets filters—improves the DW value but at the cost of computational complexity. The highest value of the DW function is obtained for the coif4 and coif5 at the 7th level of decomposition—Figure 4a. In this case, the difference of the vibration energy of the wheels is 35%. Vibration energy values are presented in Figure 4b. The threshold for detecting faulty tram wheels is marked using a dash. The MEMS-sensor acceleration samples collected during tram movement are processed and the resultant vibration energy values are compared with the thresholding parameter. The values surpassing the threshold signal faulty wheels. 

The value of the threshold for detecting faulty wheels is sensitive to the state of the rail tracks at the depot. Badly maintained rail joints or damaged sleepers generate very large vibration signals during tram movement which can deteriorate the operation of the MEMS acceleration sensors. This results in distorted values of calculated energy in the frequency bands not corresponding to the wheel states.

### 4.2. Case Study: Tram-Depot Tests and Validation

Field tests of the method were performed in the tram depot of the company Tramwaje Śląskie which provides tram services for the Upper Silesia Region in Poland. The depot maintains over 130 trams. There are several rail tracks at the depot which are used for organising the movement of trams on the premises. The chosen rail track for the tests is about 150 m long. 

A prototype of the vibration sensor is designed using a three-axis MEMS accelerometer. The measurement ranges are +/−4 g, +/−8 g, and +/−16 g. The accelerometer is AEC-Q100 qualified, it is suitable for work in harsh environments. The prototype is self-powered and capable of registering acceleration values for several weeks. In the course of a number of tests the optimal measurement range was determined. This range +/−8 g enables the registration of impacts at rail joints without “overloading” the sensor and gives large signal values during wheel rolling. Figure 5 depicts the sensor attached to the rail. A special metal bracket is used for reliable vibration transfer.

Several registration sessions were carried out with trams with wheel-sets in different technical conditions. Representative data sets for “normal” and “faulty” wheels were obtained and these are the basis for the substantiation of the idea that vibration energy is adequate for distinguishing tram wheels. Figure 6 presents excerpts from the acceleration database. These are raw data registered by the internal 12-bit A/C converter of the sensor.

The MEMS sensor is placed in the enclosure of the prototype in such a way that the *y* axis values represent lateral movements of the rail, whereas *z* axis represents the horizontal. The samples of acceleration in the *y* and *z* axis do not map the vibrations in a consistent way. These phenomena may be a consequence of the condition of the rails at the depot. The *x* axis data are chosen for determining the detection threshold for classification of the tram wheels.

### 4.3. Tram Depot Test Results Discussion

MODWPT is applied to the collected data. Coiflet with five vanishing moments and the 7th level of decomposition are used as parameters of the transform. Temperature is the factor which mostly changes the measured acceleration values as the device is factory calibrated. The sensor datasheet notes a maximal zero-g level change of +/−5 mg/°C and sensitivity change due to temperature typically +/−0.01%/°C. In all, the measurement errors do not exceed single percent values for yearly temperature changes.

Figure 7a shows the values of the DW function for conducted field tests. Randomly chosen pairs of normal and faulty wheel samples are processed and the results show that the DW function can reach values above 700% in the frequency band 418–422 Hz. Such a large range of DW values reflects the rather poor condition of the rail tracks in the depot. Taking into account all data, the largest value of DW is only 35%. The values of vibration energy—Figure 7b—are marked for the frequency band which gives the largest interval between normal and faulty tram wheels. The resultant threshold value is 2800. Error bars associated with the energy values do not surpass the threshold.

Faulty-wheel detection tests using the threshold prove that the threshold value is robust to variation in tram speed in the range of 2–7 m/s which covers the range of speeds of maneuvering trams in the tram depot. 

The calculation of the MODWPT coefficients for the 7th level of decomposition and coif5 wavelet does not impose excessive requirements on the computation resources. An embedded system incorporating the MEMS sensor and an advanced microcontroller can handle the stream of samples and calculate their energy.

The tested prototype is mounted on one rail and detects faults of wheels which roll on this rail. A complete system requires a sensor at each rail. Additionally, a collected history of sensor data assigned to trams will help maintenance works in the depot.

## 5. Conclusions

Detection of tram-wheel faults can be done successfully using commonly available MEMS-based acceleration sensors with at least a 1 kHz sampling rate. The amplitudes of registered accelerations fall within the measuring capabilities of such sensors. It is important to adapt the parameters of the detection method to the conditions of the measuring site and update them with changes. MODWPT was successfully used for the calculation of the vibration energy of the samples and does not require large computation resources. 

The tests were carried out for trams with flat wheels. The method is sensitive to different sizes of flat spots. Future studies are proposed to determine the capability of the method to assess the size of the spots and to detect other types of wheel damage. As the tram depot where the tests were conducted maintains a large fleet of trams in different technical conditions, it will be the basis for collecting vibration data. The range of wheel irregularities of such a fleet can be regarded as representative for carrying out studies. The analysis of vibration energy content in different frequency bands has proven successful for detecting wheel irregularities. The different types of irregularities are manifested by changes of vibration energy in specific frequency bands. The highest vibration frequencies, at low vehicle movement speeds, do not surpass the frequency bandwidths of commonly available MEMS sensors.

This approach may be supplemented by machine learning techniques. Machine learning, especially methods using convolutional neural networks, for classification require careful design. The design needs an extensive database for training and validation. Properly trained can be significantly more robust to changing conditions in the tram depot. This property is much desired by the management of the depot. The machine learning approach will be another goal of future studies.

## Figures and Tables

**Figure 1 sensors-22-06373-f001:**
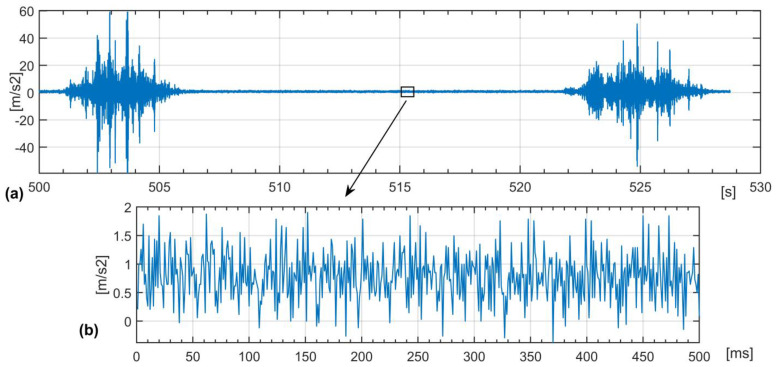
Sensor data (**a**) wheel impacts at rail joints; (**b**) rolling wheels data.

**Figure 2 sensors-22-06373-f002:**
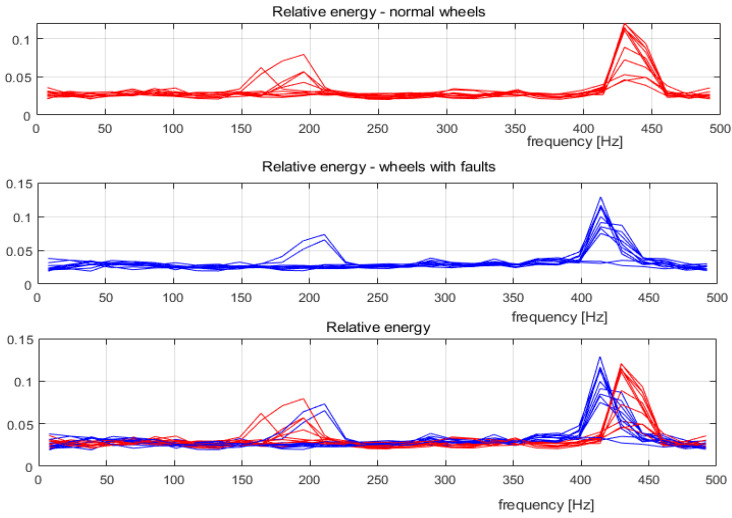
Relative energy of the sensor samples (red lines: relative energy of normal wheels; blue lines: relative energy of faulty wheels).

**Figure 3 sensors-22-06373-f003:**
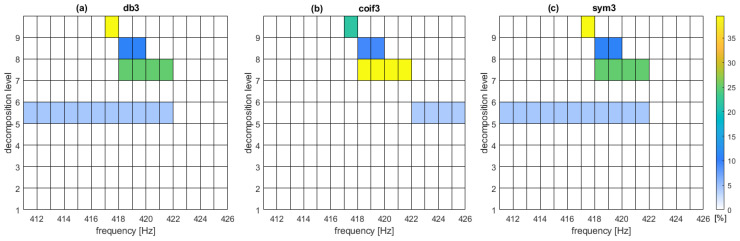
DW function values (yellow: maximum; blue: higher; green: lower and light blue: minimum values) expressed in [%]: (**a**) db3-Daubechies wavelet; (**b**) sym3-Symlet; (**c**) coif3–Coiflet.

**Figure 4 sensors-22-06373-f004:**
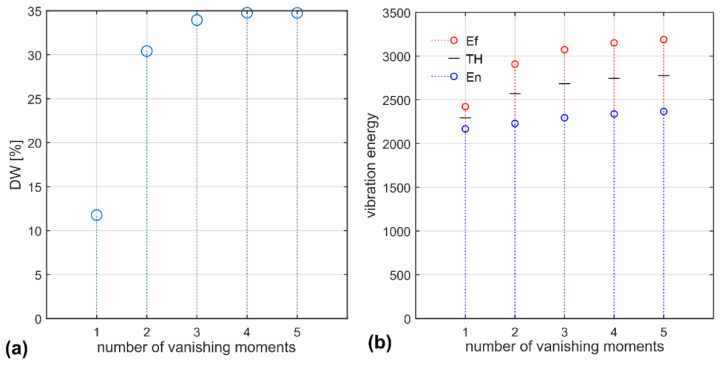
DW function values: (**a**) 7th decomposition level, Coiflets with 1–5 vanishing moments; (**b**) vibration energy calculated using coif5 wavelet (Ef-red circles: faulty wheel vibration energy; En-blue circles: normal wheel vibration energy) and threshold (TH-thresholding values of vibration energy difference between faulty and normal wheels).

**Figure 5 sensors-22-06373-f005:**
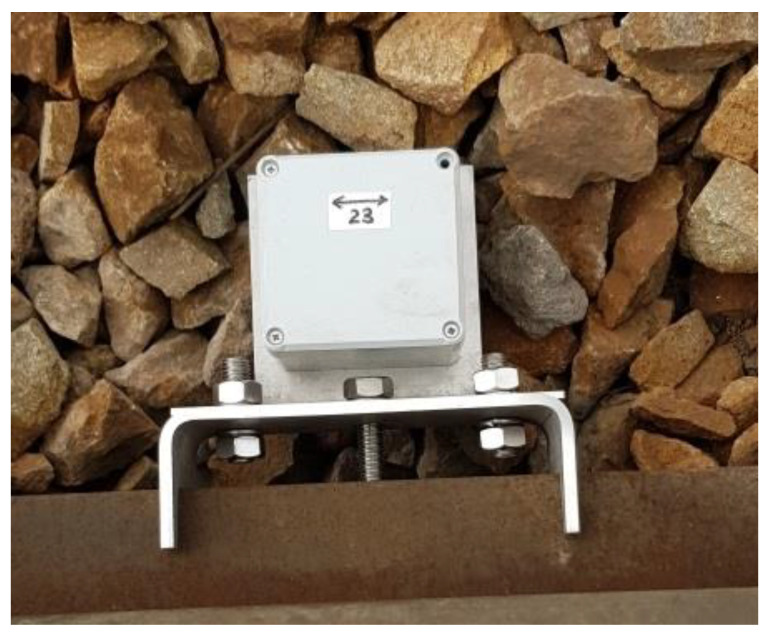
Sensor prototype mounted on rail.

**Figure 6 sensors-22-06373-f006:**
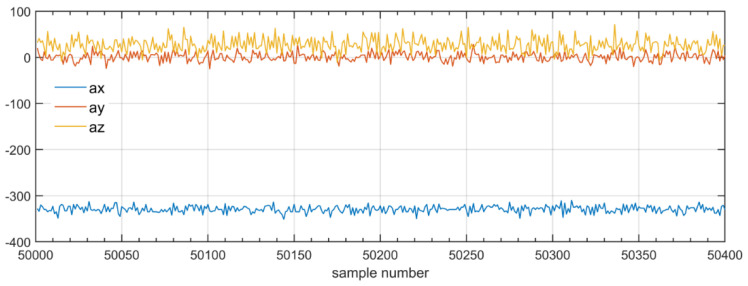
Raw acceleration data from the sensor prototype.

**Figure 7 sensors-22-06373-f007:**
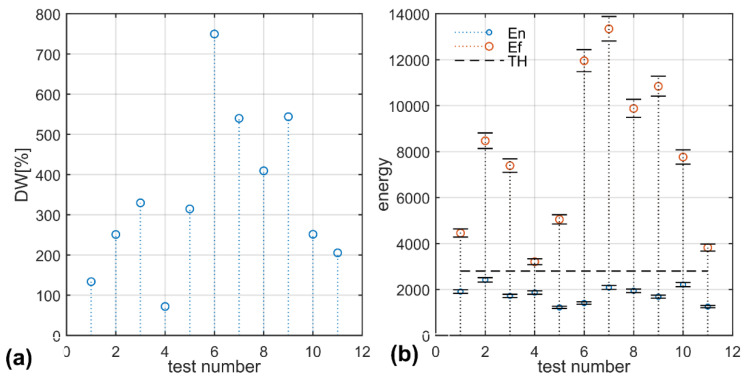
Results: (**a**) DW function (blue circles: maximum DW value for each test); (**b**) energies Ef-red circles: faulty wheel vibration energy; En-blue circles: normal wheel vibration energy) and threshold (TH-thresholding values of vibration energy difference between faulty and normal wheels).

**Table 1 sensors-22-06373-t001:** DW values for (Daubechies, Coiflets, and Symlets).

Decomposition Level	Db3	Db4	Db5	Coif3	Coif4	Coif5	Sym3	Sym4	Sym5
5	4.4	0	0.64	3.5	6.2	7.0	4.4	0	0.64
6	0.67	0	0	0	0	0	0.67	0	0
7	25	30	32	34	**35**	**35**	25	30	32
8	11	10	10	8.1	5.9	4.1	11	10	10
9	40	33	26	19	12	7.2	40	33	26

## Data Availability

Not applicable.

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
