# Peer review of "Detection of Tram Wheel Faults Using MEMS-Based Sensors"

_sensors, 2022, doi:10.3390/s22176373_

Round 1

Reviewer 1 Report

The paper is dedicated to the detection of tram wheel faults with a MEMS sensor. Adequately, the Authors present the method applied for processing the sensor readings making use of the estimation of energy of vibrations and the Maximal Overlap Discrete Wavelet Packet Transform. Additional analyses are also described that have been used for testing the functionalities and characteristics of the proposed measurement technique, including investigation on frequency bands, the energy threshold for damage detection, and the parameters of the above mentioned transform. The Authors clearly state the objective of the reported research, show the elaborated functionalities of the tram inspection system and present a comprehensive state-of-the art.

In the reviewer’s opinion the manuscript is worth to be published, however, the following comments need to be addressed.

Is the method general and can be applied for other types of vehicles, tram’s wheels, etc.? Is it sufficiently sensitive regarding a damage for other types of wheels? This should be commented in the paper.

What is the influences of the length (N) of the sensors samples set S on the capability and performance of the proposed method? Similarly, what is the impact of the length of filter L for the method performance? This issue should be commented in the paper.

What is the influence of the choice on the frequency band (the Authors state in Line 235: “Further analysis 235 is required to extract frequency bands which differentiate the vibrations caused by the 236 movement of the wheels.”) and, similarly, the sampling frequency (resolution). This issue should be commented in the paper.

Fig. 1: I believe the average value of the measured acceleration is the acceleration due to gravity – g. I would propose to exclude (subtract) this acceleration (remove the trend) to focus only on the fluctuations (changes) of the mentioned quantity and modify Figure 1.

What types of damages can be detected employing the proposed solution? How to relate the obtained results of the conducted analyses with a specific type of the experimentally identified physical damage? This issue should be commented in the paper.

Is the used sensor a commercially available 3-axis MEMS accelerometer? If so, it is recommended to add its name (IC label) to the manuscript. The description of the MEMS sensor is very limited. More details on its construction and measurement capabilities should be presented. This issue should be commented in the paper.

Figure 6: I am confused about the level of values for the X component of acceleration which is approx.. -300 (anyway, I am aware of the fact that these values are raw readings), What is the scaling factor and offset and, finally, what is a real value of the acceleration (m/s2)?

Flaws, minor issues:

Line 17: “ weighed“->”weighted”

18: “DW”->”(DW)”

187: “equivalent”->”is equivalent”

189: “enable”->”enables”

221:” joints Figure 1(b)”->” joints - Figure 1(b)”

247: full stop at the end

252: “(DW)” should be used right after the expression “weighted difference” in row 250

Author Response

Dear Sir,

thank you very much for your careful review of our paper. The comments enabled an improvement of our work, and we hope that we satisfactorily addressed the indicated deficiencies.

Keyword definition: Ad#-->Author discussion/description in the order of reviewer comments.

Ad1. In our opinion the method can be applied to detect flat wheel faults and other types of wheel surface irregularities. The idea is based on assessing the vibration energy in characteristic frequency bands. In the conclusion section we include a paragraph describing the concept of future studies of this problem.

Ad2. Thank you for this motion. Increasing the number of wavelets vanishing moments – length L of the wavelets filters, improves the DW value but at the cost of computational complexity. In order to diminish the influence of possible interference the set of samples is extended to 4096 ensuring 8 coefficients for the description of the frequency bands. The set of 4096 samples is collected in about 4 seconds which is acceptable for performing wheel diagnosis in real time. Extending the set of samples may be inadvisable because the speed of moving tram is variable, and a larger set may include the impact data.

The section 4.1 is extended to include this clarification.

Ad3. Preliminary tests show potential frequency bands which indicate some anomalies in the vibration image, and this must be studied to determine whether it is normal or it is due to wheel faults. The noted lines (235-236) are only an indication of the following clarification of the research problem.

Ad4. Thank you for detecting this shortcoming. We have corrected the figure and included a more meaningful image of the rolling data.

Ad5. Referring to ad1 carried out tests indicate that wheel faults generate higher vibration energy in the specified frequency range, but the tests were not designed to classify the faults the goal was to detect them.

Ad6. We did not apply to the manufacturer to obtain a permission for disclosing the sensor name. It is a sensor used in cars it is AEC-Q100 specified and has a measurement range +/-4g, +/-8g and +/-16g. Some more specifications are added at the end of section 4.2 and in section 4.3.

Ad7. Indeed this figure may be misleading. Our intention was to show the “image” of real data from the sensor. Calibrated acceleration values are exemplified in Figure 1. Perhaps this figure is surplus.

Ad8. Minor issues, Thank you for spotting writing errors, we corrected them.

With regards,

authors 

****************************************************************

Reviewer 2 Report

The paper is not extraordinary nevertheless it suits the aims and scope of the journal. Therefore, it can be given further consideration.

The literature review should be concluded with a research gap that will be considered in the paper.

The Authors should investigate more review papers s for example Kostrzewski, M.; Melnik, R. Condition Monitoring of Rail Transport Systems: A Bibliometric Performance Analysis and Systematic Literature Review. Sensors 2021, 21, 4710. https://doi.org/10.3390/s21144710

or Czyczuła, W., Rochel, M. (2021). Operational problems of tramway infrastructure in sharp curves. Technical Transactions: e2021015. https://doi. org/10.37705/TechTrans/e2021015

The Authors should give examples in the case of the following phrase: ”A comprehensive review of the above articles indicates that detecting wheel faults under variable-speed, loading and track conditions in a real-time is a difficult task that has only been rarely reported in the scientific literature.”

”In addition, MEMS sensors were not utilized in the earlier research for the purpose of accurately measuring the vibration signatures of passing trains, which is an essential component of condition monitoring for wheel-sets.” - The authors must investigate it more.

http://dx.doi.org/10.1016/j.proeng.2016.06.222

https://www.digikey.com/en/articles/using-a-mems-sensor-for-vibration-monitoring

https://doi.org/10.3390/s21144710 

The methodology should be given as a separate section.

The assumption and limitations of data and the fundamental research should be described.

Future research should be mentioned in the concluding section (more than ”The machine learning approach will be the goal of future studies.”)

Author Response

Dear Sir,

thank you very much for your careful review of our paper. The comments enabled an improvement of our work, and we hope that we satisfactorily addressed the indicated deficiencies.

Keyword definition: Ad#-->Author discussion/description in the order of reviewer comments.

Ad1. Thank you very much for bringing to our attention literature items which contain reports related to the papers subject. We extended section 2 and clarified the problem of vibration signatures. Section 2 is appended with conclusions defining the research gap and projected contributions of the paper in this field.

Ad2. We changed the title of section 3 to emphasize the subject of developing the method for wheel faults detection.

Ad3. Thank you for noting the lack of assumptions and limitations of data. We extend the clarification of these problems in section 4.1

Ad4. We included in section 5 clear propositions for future studies, that is work on developing the capability of the method for detecting and measuring the wheel irregularities. The studies will be done in cooperation with the tram depot where the first tests were carried out.

With regards,

Authors

****************************************************************

Reviewer 3 Report

I recommend its publication with minor revision.

Comments are listed below,

1. Introduction part is well written, but the literature citation is not strong, and the Author should include the  recent literatures,

2. English needs some improvement.

3. In the introduction part, the information of  prepared sensor n detail   need to be added in the introduction part.

4. How did you derive the calibration curve (On the basis of peak current or peak height)?

5. What is the error limit in the calculation of  the sensor 

6. Quality of all the figures is poor, author should improve with more resolution.

7. Recheck all the equations, abbreviations, place of figures and captions.

8. Author should mention  error bar 

9. Introduction needs to divided into many paragraphs.

Author Response

Dear Sir,

thank you very much for your careful review of our paper. The comments enabled an improvement of our work and we hope that we satisfactorily addressed the indicated deficiencies.

Keyword definition: Ad#-->Author discussion/description in the order of reviewer comments.

Ad1. Thank you for the detailed review of our bibliography items. We have extended section 2 to include new reports.

Ad2. A proofread was done and some errors are corrected.

Ad3. We did not apply to the manufacturer to obtain a permission for disclosing the sensor name. It is a sensor used in cars, it is AEC-Q100 specified and has a measurement range +/-4g, +/-8g and +/-16g. Some more specifications are added at the end of section 4.2 and in section 4.3. as well as in the introduction.

Ad5. Ad8. Thank you for noting this deficiency. We use the specifications of the sensor presented in the official datasheets from the manufacturer. This enabled a supplementation of the plots with error bars, we also include an explanation in section 4.3

Ad6. Unfortunately, we used the default settings of our graphing program. Thank you for such a close look at our graphs. We have corrected all our figures.

Ad7. We have rechecked the equations, abbreviations, place of figures and captions.

Ad9. We introduced shorter paragraphs in the introduction.

With regards,

Authors

****************************************************************

Round 2

Reviewer 2 Report

The Authors have developed the paper, therefore it can be considered by the Editors for publication.